# Clockwork Variational Autoencoders

**Vaibhav Saxena**
University of Toronto
vaibhav@cs.toronto.edu

**Jimmy Ba**
University of Toronto
jba@cs.toronto.edu

**Danijar Hafner**
University of Toronto
Google Research, Brain Team
mail@danijar.com

## Abstract

Deep learning has enabled algorithms to generate realistic images. However, accurately predicting long video sequences requires understanding long-term dependencies and remains an open challenge. While existing video prediction models succeed at generating sharp images, they tend to fail at accurately predicting far into the future. We introduce the Clockwork VAE (CW-VAE), a video prediction model that leverages a hierarchy of latent sequences, where higher levels tick at slower intervals. We demonstrate the benefits of both hierarchical latents and temporal abstraction on 4 diverse video prediction datasets with sequences of up to 1000 frames, where CW-VAE outperforms top video prediction models. Additionally, we propose a Minecraft benchmark for long-term video prediction. We conduct several experiments to gain insights into CW-VAE and confirm that slower levels learn to represent objects that change more slowly in the video, and faster levels learn to represent faster objects.[1]

## 1 Introduction

Video prediction involves generating high-dimensional sequences for many steps into the future. A simple approach is to predict one image after another, based on the previously observed or generated images (Oh et al., 2015; Kalchbrenner et al., 2016; Babaeizadeh et al., 2017; Denton and Fergus, 2018; Weissenborn et al., 2020). However, such temporally autoregressive models can be computationally expensive because they operate in the high-dimensional image space and at the frame rate of the dataset. In contrast, humans have the ability to reason using abstract concepts and predict their changes far into the future, without having to visualizing the details in the input space or at the input frequency.

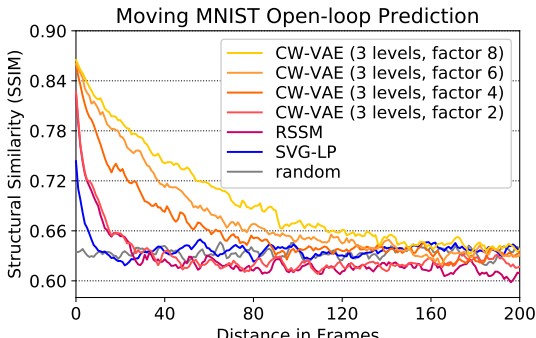

Figure 1: Video prediction quality as a function of the distance predicted. We show 4 versions of Clockwork VAE with temporal abstraction factors 2, 4, 6, and 8. Larger temporal abstraction directly results in predictions that remain accurate for longer horizons. Clockwork VAE further outperforms the top video models RSSM and SVG.

Latent dynamics models predict a sequence of learned latent states forward that is then decoded into the video, without feeding generated frames back into the model (Kalman, 1960; Krishnan et al., 2015; Karl et al., 2016). These models are typically trained using variational inference, similar to VRNN (Chung et al., 2015), except that the generated images are not fed back into the model. Latent dynamics models have recently achieved success in reinforcement learning for learning world models from pixels (Ha and Schmidhuber, 2018; Zhang et al., 2019; Buesing et al., 2018; Mirchev et al., 2018; Hafner et al., 2019b;a; 2020).

---

[1]All code is publicly available at https://github.com/vaibhavsaxena11/cwvae.

35th Conference on Neural Information Processing Systems (NeurIPS 2021).

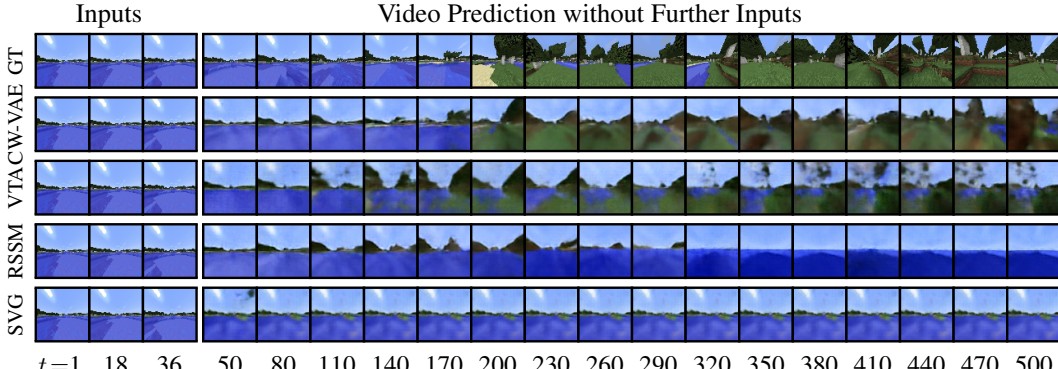

Figure 2: Long-horizon video predictions on the MineRL Navigate dataset. In the dataset, the camera moves straight ahead most of the time. CW-VAE accurately predicts the camera movement from the ocean to the forest until the end of the sequence. In contrast, VTA outputs artifacts in the sky after 240 steps and shows less diversity in the frames. RSSM fails to predict the movement toward the island after 290 frames. SVG simply copies the initial frame and does not predict any new events in the future.

To better represent complex data, hierarchical latent variable models learn multiple levels of features. Ladder VAE (Sønderby et al., 2016), VLAE (Zhao et al., 2017), NVAE (Vahdat and Kautz, 2020), and very deep VAEs (Child, 2020) have demonstrated the success of this approach for generating static images. Hierarchical latents have also been incorporated into deep video prediction models (Serban et al., 2016; Kumar et al., 2019; Castrejón et al., 2019). These models can learn to separate high-level details, such as textures, from low-level details, such as object positions. However, they operate at the frequency of the input sequence, making it challenging to predict far into the future.

Temporally abstract latent dynamics models predict learned features at a slower frequency than the input sequence. This encourages learning long-term dependencies that can result in more accurate predictions and enable computationally efficient long-horizon planning. Temporal abstraction has been studied for low-dimensional sequences (Koutník et al., 2014; Chung et al., 2016; Mujika et al., 2017). VTA (Kim et al., 2019) models videos using two levels of latent variables, where the fast states decide when the slow states should tick. TD-VAE (Gregor and Besse, 2018) models high-dimensional input sequences using jumpy predictions, without a hierarchy. Refer to Appendix B for further related work. Despite this progress, scaling temporally abstract latent dynamics to complex datasets and understanding how these models organize the information about their inputs remain open challenges.

In this paper, we introduce the Clockwork Variational Autoencoder (CW-VAE), a simple hierarchical latent dynamics model where all levels tick at different fixed clock speeds. We conduct an extensive empirical evaluation and find that CW-VAE outperforms existing video prediction models. Moreover, we conduct several experiments to gain insights into the inner workings of the learned hierarchy.

Our key contributions are summarized as follows:

- **Clockwork Variational Autoencoder (CW-VAE)**  We introduce a simple hierarchical video prediction model that leverages different clock speeds per level to learn long-term dependencies in video. A comprehensive empirical evaluation shows that on average, CW-VAE outperforms strong baselines, such as SVG-LP, VTA, and RSSM across several metrics.

- **Long-term video benchmark**  In the past, the video prediction literature has mainly focused on short-term video prediction of under 100 frames. To evaluate the ability to capture long-term dependencies, we propose using the Minecraft Navigate dataset as a challenging benchmark for video prediction of 500 frames.

- **Accurate long-term predictions**  Despite the simplicity of fixed clock speeds, CW-VAE improves over the distance of accurate video prediction of prior work. On the Minecraft Navigate dataset, CW-VAE accurately predicts for over 400 frames, whereas prior work fails before or around 150 frames.

- **Adaptation to sequence speed**  We demonstrate that CW-VAE automatically adapts to the frame rate of the dataset. Varying the frame rate of a synthetic dataset confirms that the slower latents are used more when objects in the video are slower, and faster latents are used more when the objects are faster.

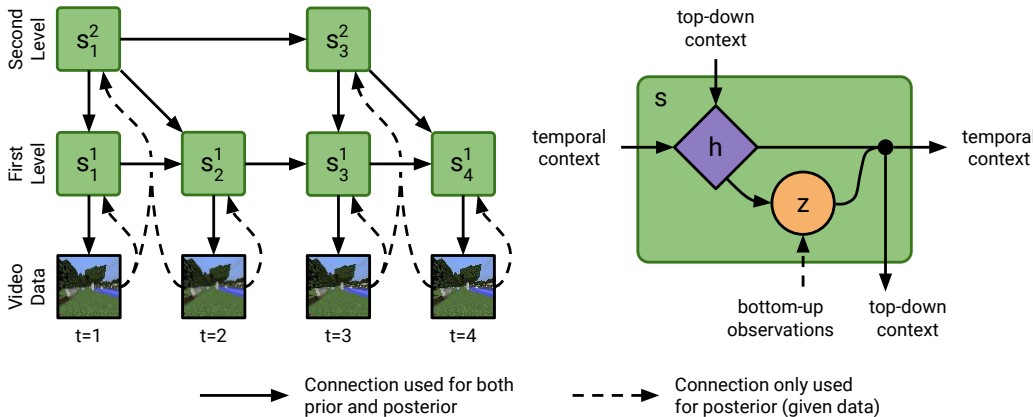

Figure 3: Clockwork Variational Autoencoder (CW-VAE). We show 2 levels with temporal abstraction factor 2 (but we use up to 4 levels and abstraction factor of 8 in our experiments). Left is the structure of the model, where each latent state $s_t^l$ in the second level conditions two latent states in the first level. The solid arrows represent the generative model, while both solid and dashed arrows comprise the inference model. On the right, we illustrate the internal components of the state, containing a deterministic variable $h_t$ and a stochastic variable $z_t$. The deterministic variable aggregates contextual information and passes it to the stochastic variable to be used for inference and stochastic prediction.

- **Separation of information**  We visualize the content represented at higher levels of the temporal hierarchy. This experiment shows that the slower levels represent content that changes more slowly in the input, such as the wall colors of a maze, while faster levels represent faster changing concepts, such as the camera position.

## 2  Clockwork Variational Autoencoder

Long video sequences contain both, information that is local to a few frames, as well as global information that is shared among many frames. Traditional video prediction models that predict ahead at the frame rate of the video can struggle to retain information long enough to learn such long-term dependencies. We introduce the Clockwork Variational Autoencoder (CW-VAE) to learn long-term correlations of videos. Our model predicts ahead on multiple time scales, as visualized in Figure 3. We build our work upon the Recurrent State-Space Model (RSSM; Hafner et al., 2019b), the details of which can be found in Appendix A.

CW-VAE consists of a hierarchy of recurrent latent variables, where each level transitions at a different clock speed. We slow down the transitions exponentially as we go up in the hierarchy, i.e. each level is slower than the level below by a factor $k$. We denote the latent state at timestep $t$ and level $l$ by $s_t^l$ and the video frames by $x_t$. We define the set of active timesteps $\mathcal{T}_l$ for each level $l \in [1, L]$ as those instances in time where the state transition generates a new latent state,

$$\text{Active steps:} \quad \mathcal{T}_l \doteq \{t \in [1, T] \mid t \bmod k^{l-1} = 1\}. \tag{1}$$

At each level, we condition $k$ consecutive latent states on a single latent variable in the level above. For example, in the model shown in Figure 3 with $k = 2$, $\mathcal{T}_1 = \{1, 2, 3, \dots\}$, $\mathcal{T}_2 = \{1, 3, 5, \dots\}$, and both $s_1^1$ and $s_2^1$ are conditioned on the same $s_1^2$ from the second level.

The latent chains can also be thought of as a hierarchy of latent variables where each level has a separate state variable per timestep, but only updates the previous state every $k^{l-1}$ timesteps and otherwise copies the previous state, so that $\forall t \notin \mathcal{T}_l$:

$$\text{Copied states:} \quad s_t^l \doteq s_{\max_{\tau}\{\tau \in \mathcal{T}_l \mid \tau \leq t\}}^l. \tag{2}$$

**Joint distribution**  We can factorize the joint distribution of a sequence of images and active latents at every level into two terms: (1) the reconstruction terms of the images given their lowest level latents, and (2) state transitions at all levels conditioned on the previous latent and the latent above,

$$p(x_{1:T}, s_{1:T}^{1:L}) \doteq \left( \prod_{t=1}^T p(x_t \mid s_t^1) \right) \left( \prod_{l=1}^L \prod_{t \in \mathcal{T}_l} p(s_t^l \mid s_{t-1}^l, s_t^{l+1}) \right). \tag{3}$$

To implement this distribution and its inference model, CW-VAE utilizes the following components, $\forall\, l \in [1, L], t \in \mathcal{T}_l$,

$$
\begin{aligned}
&\text{Encoder:} && e_t^l = e(x_{t:t+k^{l-1}-1}) \\
&\text{Posterior transition } q_t^l: && q(s_t^l \mid s_{t-1}^l, s_t^{l+1}, e_t^l) \\
&\text{Prior transition } p_t^l: && p(s_t^l \mid s_{t-1}^l, s_t^{l+1}) \\
&\text{Decoder:} && p(x_t \mid s_t^1).
\end{aligned}
\tag{4}
$$

**Inference**  CW-VAE embeds the observed frames using a CNN. Each active latent state at a level $l$ receives the image embeddings of its corresponding $k^{l-1}$ observation frames (dashed lines in Figure 3). The diagonal Gaussian belief $q_t^l$ is then computed as a function of the input features, the posterior sample at the previous step, and the posterior sample above (solid lines in Figure 3). We reuse all weights of the generative model for inference except for the output layer that predicts the mean and variance.

**Generation**  The diagonal Gaussian prior $p_t^l$ is computed by applying the transition function from the latent state at the previous timestep in the current level, as well as the state belief at the level above (solid lines in Figure 3). Finally, the posterior samples at the lowest level are decoded into images using a transposed CNN.

**Training objective**  Because we cannot compute the likelihood of the training data under the model in closed form, we use the ELBO as our training objective. This training objective optimizes a reconstruction loss at the lowest level, and a KL regularizer at every level in the hierarchy summed across active timesteps,

$$
\max_{e,q,p} \sum_{t=1}^{T} \mathrm{E}_{q_t^1}[\ln p(x_t \mid s_t^1)] - \sum_{l=1}^{L} \sum_{t \in T_l} \mathrm{E}_{q_{t-1}^l q_t^{l+1}}\Big[\, \mathrm{KL}[q_t^l \parallel p_t^l]\Big].
\tag{5}
$$

The KL regularizers limit the amount of information about the images that enters via the encoder. This encourages the model to utilize the "free" information from the previous and above latent and only attend to the input image to the extent necessary. Since the number of active timesteps decreases as we go higher in the hierarchy, the number of KL terms per level decreases as well. Hence it is easier for the model to store slowly changing information high up in the hierarchy than to pay a KL penalty to repeatedly extracting the information from the images at the lower level or trying to remember it by passing it along for many steps at the lower level without accidental forgetting.

**Stochastic and Deterministic Path**  As shown in Figure 3 (right), we split the state $s_t^l$ into stochastic ($z_t^l$) and deterministic ($h_t^l$) parts (Hafner et al., 2019b). The deterministic state is computed using the top-down and temporal context, which then conditions the stochastic state at that level. The stochastic variables follow diagonal Gaussians with predicted means and variances. We use one GRU (Cho et al., 2014) per level to update the deterministic variable at every active step. All components of Equation 4 jointly optimize Equation 5 by stochastic backprop with reparameterized sampling (Kingma and Welling, 2013; Rezende et al., 2014). Refer to Appendix C for architecture details.

## 3  Experiments

We compare the Clockwork Variational Autoencoder (CW-VAE) on 4 diverse video prediction datasets to state-of-the-art video prediction models in Sections 3.2 to 3.5. We then conduct an extensive analysis to gain insights into the behavior of CW-VAE. We study the content stored at each level of the hierarchy in Section 3.6, the effect of different clock speeds in Section 3.7, the amount of information at each level in Section 3.8, and the change in information as a function of the dataset frame rate Section 3.9. The source code and video predictions are available on the project website.[2]

**Datasets**  We choose 4 diverse video datasets for the benchmark. The MineRL Navigate dataset (available under the CC Attribution-NonCommercial-ShareAlike 4.0 license) was crowd sourced by Guss et al. (2019) for reinforcement learning applications. We process this data to create a long-horizon video prediction dataset that contains ~750k frames. The sequences show players traveling to goal locations in procedurally generated 3D worlds of the video game Minecraft (traversing forests, mountains, villages, oceans). The KTH Action video prediction dataset (Schuldt et al., 2004) (available under the CC Attribution-NonCommercial license) contains 290k frames. The 600 videos show humans walking, jogging, running, boxing, hand-waving, and clapping. GQN Mazes (Eslami

---

[2] https://danijar.com/cwvae

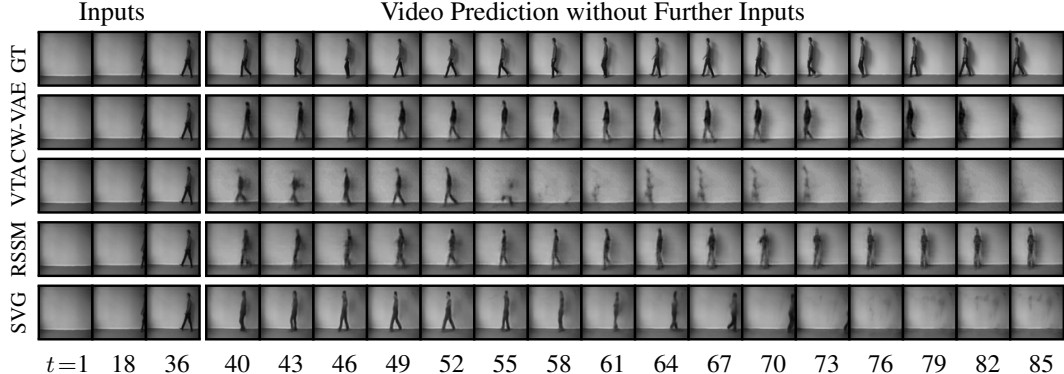

Figure 4: Open-loop video predictions for KTH Action. CW-VAE predicts accurately for 50 time steps. VTA fails to predict coherent transitions from one frame to the next around time step 55, which we attribute to the fact that its lower level is reset when the higher level steps. SVG and RSSM accurately predict for 20 frames and their subsequent predictions appear plausible but fail to capture the long-term dependencies in the walking movement that is present in the dataset.

et al., 2018) (available under the Apache License 2.0) contains 9M frames of videos of a scripted policy that traverses procedurally generated mazes with randomized wall and floor textures. For Moving MINST (Srivastava et al., 2015) (available under the CC Attribution-ShareAlike 3.0 license) we generate 2M frames where two digits move with velocities sampled uniformly in the range of 2 to 6 pixels per frame and bounce within the edges.

**Baselines** We compare CW-VAE to 3 well-established video prediction models, and an ablation of our method where all levels tick at the fastest scale, which we call NoTmpAbs. VTA (Kim et al., 2019) is the state-of-the-art for video prediction using temporal abstraction. It consists of two levels, with the lower level predicting when the higher level should step. The lower level is reset when the higher level steps. RSSM (Hafner et al., 2019b) is commonly used as a world model in reinforcement learning. It predicts forward using a sequence of compact latents without temporal abstraction. SVG-LP (Denton and Fergus, 2018), or SVG for short, has been shown to generate sharp predictions on visually complex datasets. It autoregressively feeds generated images back into the model while also using a latent variable at every time step. The parameter counts are shown in Table D.2. NoTmpAbs simply sets the temporal abstraction factor to 1 and thus uses the same number of parameters as its temporally-abstract counterparts.

**Training details** We train all models on all datasets for 300 epochs on training sequences of 100 frames of size $64 \times 64$ pixels. For the baselines, we tune the learning rate in the range $[10^{-4}, 10^{-3}]$ and the decoder stddev in the range $[0.1, 1]$. We use a temporal abstraction factor of 6 per level for CW-VAE, unless stated otherwise. Refer to Appendix D for hyperparameters and training durations. A 3-level CW-VAE with abstraction factor 6 takes 2.5 days to train on one Nvidia Titan Xp GPU. Higher temporal abstraction factors train faster because fewer state transitions need to be computed.

**Evaluation** We evaluate the open-loop video predictions under 4 metrics: Structural Similarity index (SSIM, higher is better), Peak Signal-to-Noise Ratio (PSNR, higher is better), Learned Perceptual Image Patch Similarity (LPIPS, lower is better; Zhang et al., 2018), and Frechet Video Distance (FVD, lower is better; Unterthiner et al., 2018). All video predictions are open-loop, meaning that the models only receive the first 36 frames as context input and then predict forward without access to intermediate frames. The number of input frames equals one step of the slowest level of CW-VAE for simplicity; see Appendix D.

## 3.1 Benchmark Scores

We evaluate the 5 models on 4 datasets and 4 metrics. To aggregate the scores, we compute how each model ranks compared to the other models, and average its ranks across datasets and metrics, shown in Figure 5c. Individual scores are shown in Table E.1.

Averaged over datasets and metrics, CW-VAE substantially outperforms the existing methods, which we attribute to its hierarchical latents and temporal abstraction. The second best model is NoTmpAbs, which uses hierarchical latents but no temporal abstraction. It is followed by RSSM, which has only one level. The performance improvement due to hierarchical latents is smaller than the additional

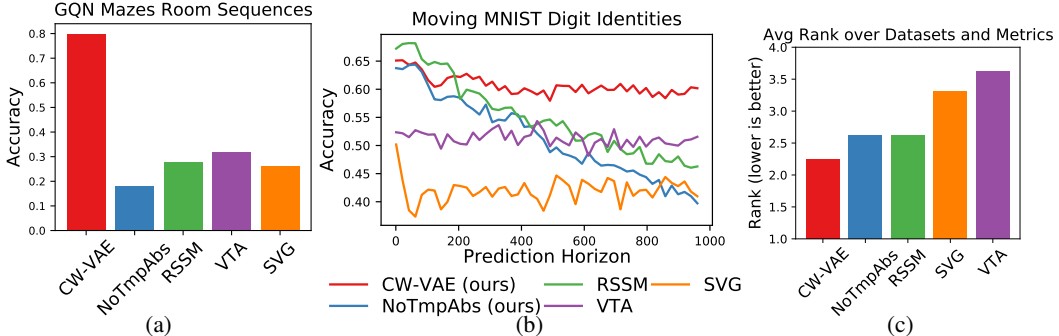

Figure 5: Evaluating video prediction models. (a) Accuracy of sequence of rooms in GQN Mazes. CW-VAE predicts video with a significantly higher accuracy of high-level details than the second best video prediction model for this evaluation (VTA). (b) Accuracy of digit identities in two-digit Moving MNIST. CW-VAE generates video with a significantly higher accuracy of digit identities until 1000 timesteps. (c) Aggregated performance ranks of all methods across 4 datasets and 4 metrics. The best possible rank is 1 and the worst is 5. Our CW-VAE uses a temporally abstract hierarchy and substantially outperforms a hierarchy without temporal abstraction (NoTmpAbs), as well as the top single-level model RSSM, the image-space model SVG, and the temporally abstract model VTA.

improvement due to temporal abstraction. Specifically, NoTmpAbs is sometimes outperformed by RSSM but CW-VAE matches or outperforms RSSM on all datasets and metrics.

While SVG achieves high scores on GQN Mazes due to its sharp predictions, it performs poorly compared to CW-VAE and RSSM on the other 3 datasets. This result indicates the benefits of predicting forward purely in latent space instead of feeding generated images back into the model. While VTA makes reasonable predictions, it lags behind the other methods in our experiments. One reason could be that its low level latents do not carry over the recurrent state when the high level ticks, which we found to cause sudden jumps in the predicted video sequence.

### 3.2 MineRL Navigate

Video predictions for sequences of 500 frames are shown in Figure 2. Given only an island on the horizon as context, the temporally abstract models CW-VAE and VTA correctly predict that the player will enter the island (as the player typically navigates straight ahead in the dataset). However, the predictions of VTA lack diversity and contain artifacts. In contrast, CW-VAE predicts diverse variations in the terrain, such as grass, rocks, trees, and a pond. RSSM generates plausible images but fails to capture long-term dependencies, such as the consistent movement toward the island. SVG only predicts small to no changes after the first few frames. Additional samples in Figure F.1 further confirm the findings. The predictions of all models are a bit blurry, which we attribute to the model capacity, which was restricted due to our limited computational resources.

### 3.3 KTH Action

Figure 4 shows video prediction samples for the walking task of KTH Action. CW-VAE accurately predicts the motion of the person as they walk across the frame until walking out of the frame. VTA predicts that the person suddenly disappears, which we attribute to it resetting the low level every time the high level steps. Both RSSM and SVG tend to forget the task demonstrated in the context frames, with RSSM predicting a person standing still, and SVG predicting the person starting to move in the opposite direction, after open-loop generation of about 20 frames. We also point out that SVG required the slower VGG architecture for KTH, whereas CW-VAE works well even with the smaller DCGAN architecture.

### 3.4 GQN Mazes

Figure G.1 shows open-loop video prediction samples of GQN Mazes, where the camera traverses two rooms connected by a hallway with distinct randomized textures in a 3D maze. CW-VAE maintains and predicts wall and floor patterns of both rooms for 200 frames, whereas RSSM and VTA fail to predict the transition to the textures of the second room. SVG is the model with the sharpest predictions on this dataset which is reflected by the metrics. However, it confuses textures over a

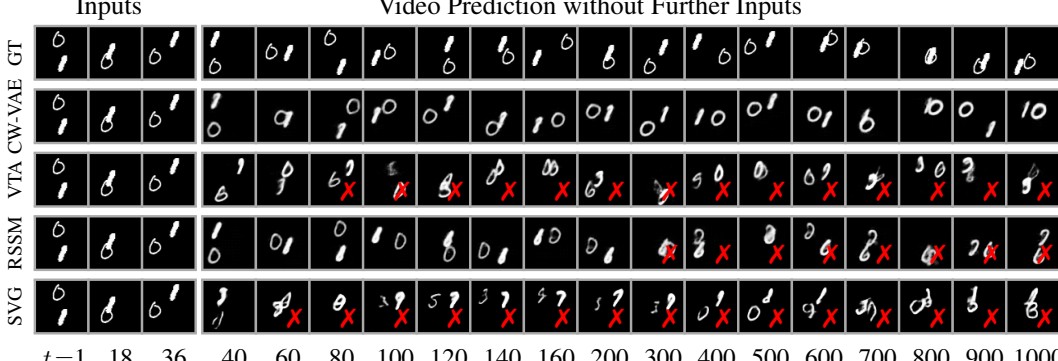

Figure 6: Long-horizon open-loop video predictions on Moving MNIST. We compare samples of CW-VAE, VTA, RSSM, and SVG. Red crosses indicate predicted frames that show incorrect digit identities. We observe that CW-VAE remembers the digit identities across all 1000 frames of the prediction horizon, whereas all other models forget the digit identities before or around 300 steps. The image-space model SVG is the first to forget the digit identities, supporting the hypothesis that prediction errors accumulate more quickly when predicted images are fed back into the model compared to predicting forward purely in latent space.

long horizon, generating mixed features on the walls and floor of the maze as visible in the video predictions in Figure G.1. While the open-loop predictions of CW-VAE differ from the ground truth in their camera viewpoints, the model remembers the wall and floor patterns for all 200 timesteps, highlighting its ability to maintain global information for a longer duration.

To better evaluate CW-VAE, we compute the prediction accuracy for the following three high-level categories of sequences of rooms in the ground-truth video: 1) agent moves across the same room throughout the video, 2) agent traverses into the hallway but does not transition into another room, 3) agent goes into the hallway and traverses back into a room. Because the rooms and hallways use different textures, we can easily identify these classes via the color histogram. Figure 5a shows that CW-VAE predicts video with a significantly higher accuracy than the second best model, VTA, which is then followed by other baselines with a similar accuracy. This shows that CW-VAE immensely benefits from its temporal hierarchy for accurately predicting high-level details such as the sequence of rooms in a video.

### 3.5 Moving MNIST

Figure 6 shows samples of open-loop video prediction of 1000 frames. We observe that CW-VAE remembers the digit identity for all 1000 timesteps. RSSM clearly outperforms SVG, which typically forgets digit identity within 50 timesteps, but starts to forget object identities much sooner than CW-VAE. Figure 5b shows the two-digit classification accuracy of video prediction as it varies over time, averaged over the test set. We observe that while RSSM initially has the highest accuracy, it falls below CW-VAE after 200 frames. CW-VAE maintains a stable accuracy over 1000 timesteps, ending with significantly more accurate predictions at the horizon than the second best model, VTA. With regards to digit positions, as shown in Figure 6, CW-VAE predicts accurate positions until 100 steps, and predicts a plausible sequence thereafter. RSSM also predicts the correct location of digits for at least as long as CW-VAE, whereas SVG starts to lose track of positions much sooner. We note

| Model | Level 1 | Level 2 | Level 3 | Level 4 |
|---|---|---|---|---|
| 4-level CW-VAE | 397.10 | 41.20 | 2.66 | 0.0001 |
| 3-level CW-VAE | 366.70 | 45.28 | 6.29 | – |
| 2-level CW-VAE | 389.40 | 39.18 | – | – |
| 1-level CW-VAE | 440.50 | – | – | – |

Table 1: KL loss at each level of the hierarchy for CW-VAEs with different numbers of levels, summed over time steps of a training sequence. Deeper hierarchies incorporate less information about the inputs into the lowest level and instead distribute the information content across levels. Higher levels tend to store less information than lower levels, suggesting that the dataset contains more short-term dependencies than long-term dependencies.

Inputs                                    Ground truth

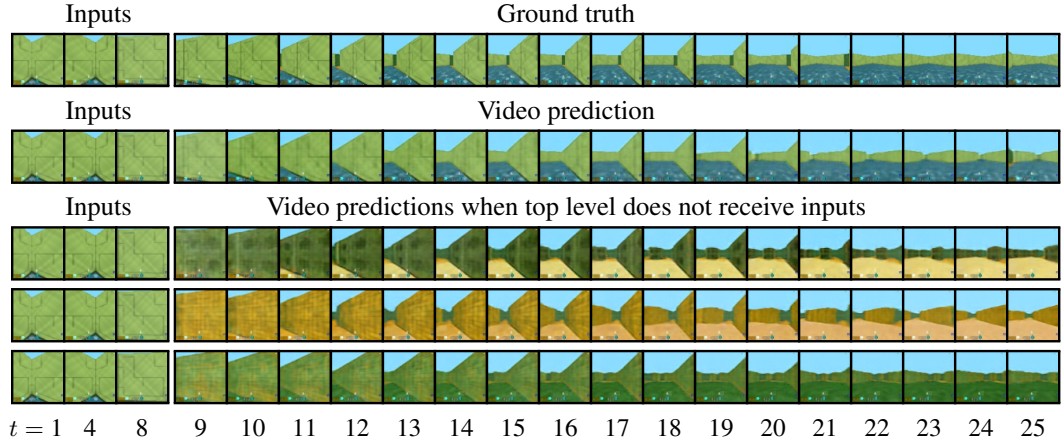

Inputs                                    Video prediction

Inputs          Video predictions when top level does not receive inputs

$t = 1$  4    8    9    10   11   12   13   14   15   16   17   18   19   20   21   22   23   24   25

Figure 7: Visualization of the content stored at the top-level of a Clockwork VAE trained on GQN Mazes. The first row shows ground truth and the second row shows a normal video prediction. The remaining rows show video predictions where the top-level stochastic variables are drawn from the prior rather than the posterior. In other words, only the lower and middle levels have access to the context images but the top level is blind. We find that the positions of camera and nearby walls remain unchanged, so this information must have been represented at lower and middle levels. In contrast, the model predicts textures that differ from the context, meaning that this global information must have been stored at the top level.

that the generations by models that predict purely in latent space are slightly blurry compared to those generated by SVG.

### 3.6 Content Stored at Different Levels

We visualize the information stored at different levels of the hierarchy. We generate video predictions where only the lower and middle layer have access to the context images, but the high level follows its prior. This way, the prediction will only be consistent with the context inputs for information that was extracted by the lower and middle level. Information that is held by the top level is not informed by the context frames and thus will follow the training distribution. We use 8 instead of 36 input images for this experiment because we do not need to compute the top level posterior here. Figure 7 shows these samples for one input sequence and more examples are included in Figure G.1. We find that the video predictions correctly continue the positions of the camera and nearby walls visible in the input frames,

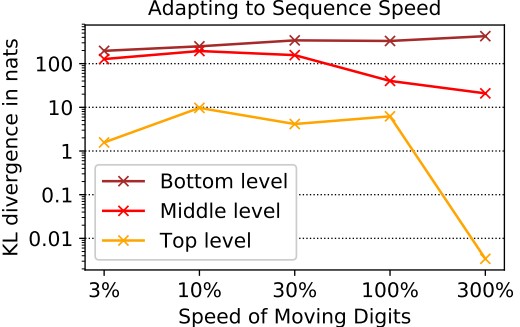

Figure 8: KL loss at each level of CW-VAE when trained on slower or faster variants of Moving MNIST. The KL value at a level indicates the amount of information stored at that level. Faster moving digits result in higher KLs at the lower level and lower KLs at higher levels. Slower moving digits result in more information at higher levels.

but that the textures appear randomized. This confirms that the top level stores the global information about the textures whereas more ephemeral information is stored in the faster ticking lower and middle levels. We also experimented with resetting the lower or middle level but found that they store similar information, suggesting that a 2-level model may be sufficient for this dataset.

### 3.7 Temporal Abstraction Factor

Figure 1 compares the quality of open-loop video predictions on Moving MNIST for CW-VAE with temporal abstraction factors 2, 4, 6, and 8 - all with equal number of model parameters. Increasing the temporal abstraction factor directly increases the duration for which the predicted frames are accurate for. Comparing to RSSM and SVG, CW-VAEs model long-term dependencies for $6\times$ as many frames as the baselines before losing temporal context. The point at which the models lose temporal context is when they approach the "random" line, which shows the quality of using randomly sampled training images as a native baseline for video prediction that has no temporal dependencies.

## 3.8 Information Amount per Level

Table 1 shows the KL regularizers for each level for CW-VAEs of varying number of levels. The KL regularizers are summed across evaluation sequences on the Moving MNIST dataset. The KL regularizers provide an upper bound on the amount of information incorporated into each level of the hierarchy, which we use as an indicator. The 2-level and 3-level CW-VAEs were trained with a temporal abstraction factor of 6, and the 4-level model with a factor of 4 to fit into GPU memory. We observe that the amount of information stored at the lowest level decreases as we use a deeper hierarchy, and further decreases for larger temporal abstraction factors. Using a larger number of levels means that the lowest level does not need to capture as much information. Moreover, increasing the amount of temporal abstraction makes the higher levels more useful to the model, again reducing the amount of information that needs to be incorporated at the lowest level.

## 3.9 Adapting to Sequence Speed

To understand how our model adapts to changing temporal correlations in the dataset, we train CW-VAE on slower and faster versions of moving MNIST. Figure 8 shows the KL divergence summed across the active timesteps of each level in the hierarchy. The KL regularizer at each level correlates with the frame rate of the dataset. The faster the digits move, the more the fast ticking lowest level of the hierarchy is used. The slower the digits move, the more the middle and top level are used. Even though the KL divergence at the top level is small, it follows a consistent trend. We conjecture that the top level stores relatively little information because the only global information is the two digit identities, which take about 5 nats to store. The experiment shows that the amount of information stored at any temporally abstract level adapts to the speed of the sequence, with high-frequency details pushed into fast latents.

# 4 Discussion

This paper introduces the Clockwork Variational Autoencoder (CW-VAE) that leverages a temporally abstract hierarchy of latent variables for long-term video prediction. We demonstrate its empirical performance on 4 diverse video prediction datasets with up to 1000 frames and show that CW-VAE outperforms top video prediction models from the literature. Moreover, we confirm experimentally that the slower ticking higher levels of the hierarchy learn to represent content that changes more slowly in the video, whereas lower levels hold faster changing information.

We point out the following limitations of our work as promising directions for future work:

- The typical video prediction metrics are not ideal at capturing the quality of video predictions, especially for long horizons. To this end, we have experimented with evaluating the best out of 100 samples for each evaluation video but have not found significant differences in the results, while evaluating even more samples is computationally infeasible to us. Using datasets where underlying attributes of the scene are available would allow evaluating the multi-step predictions by how well underlying attributes can be extracted from the representations using a separately trained readout network.

- In our experiments, we train the Clockwork VAE end-to-end on training sequences of length 100. With 3 levels and a temporal abstraction factor of 8, the top level can only step once within each training sequence. Our experiments show clear benefits of the temporally abstract model, but we conjecture that its performance could be further improved by training the top level on more consecutive transitions.

- We used relatively small convolutional neural networks for encoding and decoding the images. This results in a relatively light-weight model that can easily be trained on a single GPU in a few days. However, we conjecture that using a larger architecture could increase the quality of generated images, resulting in a model that excels at predicting both high frequency details and long-term dependencies in the data.

Temporally abstract latent hierarchies are an intuitive approach for processing high-dimensional input sequences. Besides video prediction, we hope that our findings can help advance other domains that deal with high-dimensional sequences, such as representation learning, video understanding, and reinforcement learning.

**Acknowledgments** We thank Ruben Villegas and our anonymous reviewers for their valuable input.

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
