# A Background

We build our work upon the Recurrent State-Space Model (RSSM; Hafner et al., 2019b) that has been shown to successfully learn environment dynamics from raw pixels in reinforcement learning. This model acts as an important baseline for our evaluation. RSSM explains the video sequence $x_{1:T}$ using a latent sequence of compact states $s_{1:T}$. Importantly, the model is autoregressive in latent space but not in image space, allowing us to predict into the future without generating images along the way,

$$p(x_{1:T}, s_{1:T}) \doteq \prod_{t=1}^{T} p(x_t \mid s_t) p(s_t \mid s_{t-1}). \tag{6}$$

Given a training sequence, RSSM first individually embeds the frames using a CNN. A recurrent network with deterministic and stochastic components then summarizes the image embeddings. The stochastic component help with modeling multiple futures, while the deterministic state helps remember information over many timesteps without being erased by noise. Finally, the states are decoded using a transposed CNN to provide a training signal,

$$
\begin{aligned}
&\text{Encoder:} && e_t = \text{enc}(x_t) \\
&\text{Posterior transition } q_t\text{:} && q(s_t \mid s_{t-1}, e_t) \\
&\text{Prior transition } p_t\text{:} && p(s_t \mid s_{t-1}) \\
&\text{Decoder:} && p(x_t \mid s_t).
\end{aligned} \tag{7}
$$

The posterior and prior transition models share the same recurrent model. The difference is that the posterior incorporates images, while the prior tries to predict ahead without knowing the corresponding images. The prior lets us to predict ahead purely in latent space at inference time.

As is typical with deep latent variable models, we cannot compute the likelihood of the training data under the model in closed form. Instead, we use the evidence lower bound (ELBO) as training objective (Hinton and Van Camp, 1993). The ELBO encourages to accurately reconstruct each image from its corresponding latent state, while regularizing the latent state distributions to stay close to the prior dynamics,

$$\max_{q,p} \sum_{t=1}^{T} \text{E}_{q_t}[\ln p(x_t \mid s_t)] - \sum_{t=1}^{T} \text{E}_{q_{t-1}}\Big[ \text{KL}[q_t \parallel p_t] \Big]. \tag{8}$$

The KL regularizer limits the amount of information that the posterior incorporates into the latent state at each time step, thus encouraging the model to mostly rely on information from past time steps and only extract information from each image that cannot be predicted from the preceding images already. All components jointly optimize Equation 8 using stochastic backpropagation with reparameterized sampling (Kingma and Welling, 2013; Rezende et al., 2014).

# B Additional Related Work

**Video prediction**  Video prediction has seen great progress due to deep learning (Srivastava et al., 2015; Oh et al., 2015; Vondrick et al., 2016; Lotter et al., 2016; Gemici et al., 2017). The models learn the spatial and temporal dependencies between pixels typically via autoregressive prediction or latent variables. VPN (Kalchbrenner et al., 2016) and more recently Video Transformers (Weissenborn et al., 2020) are fully autoregressive in space and time, resulting in powerful but computationally expensive models. SV2P (Babaeizadeh et al., 2017) consists of a stack of ConvLSTMs with skip connections between the encoder and decoder. It is autoregressive in time and uses a latent variable at each time step to capture dependencies between pixels, so the pixels of each image can be predicted in parallel. SVG-LP (Denton and Fergus, 2018) also combines latent variables with autoregressive prediction in time and additionally learns the latent prior. SAVP (Lee et al., 2018) uses an adversarial loss to encourage sharper predictions. DVD-GAN (Clark et al., 2019) completely relies on adversarial discriminators and scales to visually complex datasets.

Most of these video prediction models are autoregressive in time, requiring them to feed predicted images back in during prediction. This can be computationally expensive and lead to a distributional shift, because during training the input images come from the dataset (Bengio et al., 2015).

**Latent dynamics** Latent dynamics models learn all pixel dependencies via latent variables and without autoregressive conditioning, allowing them to efficiently predict forward without feeding generated images back into the model. Early examples are Kalman filters (Kalman, 1960) and hidden Markov models (Bourlard and Morgan, 1994; Bengio, 1996).

Deep learning has enabled expressive latent dynamics models. Deep Kalman filters (Krishnan et al., 2015) and deep variational Bayesian filters (Karl et al., 2016) are extensions of VAEs to sequences and have been applied to simple synthetic videos and 3D RL environments (Mirchev et al., 2018). RSSM was introduced as part of the PlaNet agent (Hafner et al., 2019b) and has enabled successful planning for control tasks (Hafner et al., 2019a) and Atari games hafner2020dreamerv2. SLRVP (Franceschi et al., 2020) develop a latent dynamics model that predicts natural video of robotic object interactions and movement of persons. Video Flow (Kumar et al., 2019) learns representations via normalizing flows instead of variational inference and have shown benefits on videos of robotic object interactions.

**Hierarchical latents** Deep learning rests on the power of hierarchical representations. Hierarchical latent variable models infer stochastic representations and learn how to generate high-dimensional inputs from them. Ladder VAE (Sønderby et al., 2016) and VLAE (Zhao et al., 2017) are deep VAEs that generate static images using a hierarchy of conditional latent variables. The recent NVAE (Vahdat and Kautz, 2020) generates high-quality natural images and Child (2020) shows that hierarchical VAEs with enough levels can outperform pixel-autoregressive models for modeling high-resolution images.

For video prediction, Wichers et al. (2018) learn hierarchical features and show that this facilitates long-term prediction. HVRNN (Castrejón et al., 2019) combines temporally autoregressive conditioning with a hierarchy of latent sequences. Video Flow (Kumar et al., 2019) learns a hierarchy of latent representations using normalizing flows and models their dynamics. These models typically still operate at the input frequency, which can make it challenging to learn long-term dependencies and creates opportunities for accumulating errors.

**Temporal abstraction** Learning long-term dependencies is a key challenge in sequence modeling. Most approached to it define representations that are rarely or slowly changing over time. For example, the recurrent state of LSTM (Hochreiter and Schmidhuber, 1997) and GRU (Cho et al., 2014) is gated to encourage remembering past information. Transformers (Vaswani et al., 2017) effectively use the concatenation of all previous representations, which are copied without being changed, as their state. Memory-augmented RNNs (Graves et al., 2014; 2016; Gemici et al., 2017) read from and write to a fixed number of slots that are copied between time steps without changes.

In contrast, models with explicit temporal abstraction allow predicting future inputs or their features more efficiently, without having to predict all intermediate steps. Clockwork RNNs (Koutník et al., 2014) partition the recurrent units of an RNN into groups that operate at different clock speeds, which inspired our paper that combines this idea with the use of latent variables. In multi-scale RNNs (Chung et al., 2016), faster features predict when to hand control to the slower features. CHiVE (Kenter et al., 2019) is a temporally abstract hierarchy for speech synthesis. Clockwork FCN (Shelhamer et al., 2016) accelerates video segmentation using a temporal hierarchy of convolutional networks.

The majority of the video prediction literature focuses on short video of up to 50 frames and relatively few models have incorporated explicit temporal abstraction. VTA (Kim et al., 2019) uses two RSSM sequences for video prediction, where the fast level predicts when to hand control over to the slow level. TD-VAE (Gregor and Besse, 2018) learns a transition function that can predict for a variable number of time steps in a single forward pass. TAP (Jayaraman et al., 2018) predicts only predictable video frames. GCP (Pertsch et al., 2020) predicts mid-points between two frames to allow dynamic sub-division.

To spur progress in this space, we propose a Minecraft dataset with sequences of 500 frames and introduce CW-VAE, a single yet successful video prediction model and shed light on its properties and representations.

## C   Model Architectures

We use convolutional frame encoders and decoders, with architectures very similar to the DCGAN (Radford et al., 2016) discriminator and generator, respectively. To obtain the input embeddings $e_t^l$ at

a particular level, $k^{l-1}$ input embeddings are pre-processed using a feed-forward network and then summed to obtain a single embedding. We do not use any skip connections between the encoder and decoder, which would bypass the latent states.

## D  Hyper Parameters

We keep the output size of the encoder at each level of CW-VAE as $|e_t^l| = 1024$, that of the stochastic states as $|p_t^l| = |q_t^l| = 20$, and that of the deterministic states as $|h_t^l| = 200$. All hidden layers inside the cell, both for prior and posterior transition, are set to $200$. We increase state sizes to $|p_t^l| = |q_t^l| = 100$, $|h_t^l| = 800$, and hidden layers inside the cell to $800$ for the MineRL Navigate dataset. We train using the Adam optimizer (Kingma and Ba, 2014) with a learning rate of $3 \times 10^{-4}$ and $\epsilon = 10^{-4}$ using a batch size of 100 sequence with 100 frames each.

We use 36 context frames for video predictions, which is the minimum number of frames required to transition at least once in the highest level of the hierarchy. With 3 levels in the hierarchy and a temporal abstraction factor of 6, each latent state at the highest level corresponds to 36 images in the sequence, and thus its encoder network expects 36 images as input.

| Model | Training time |
|---|---|
| CW-VAE (3 levels) | 20 hours |
| NoTmpAbs (3 levels) | 31 hours |
| RSSM | 18 hours |
| VTA | 21 hours |
| SVG | 37 hours |

Table D.1: Training time of the models used in the experiments for 100 epochs on one Nvidia Titan Xp GPU.

| Model | Small Version | Large Version |
|---|---|---|
| CW-VAE (3 levels) | 12M | 34M |
| NoTmpAbs (3 levels) | 12M | 34M |
| RSSM | 5M | 13M |
| SVG | 13M | 23M |
| VTA | 3M | – |

Table D.2: Total number of trainable parameters of the video prediction models used in the experiments of this paper.

# E    Quantitative Evaluation for Video Prediction

| | MineRL Navigate | | | | KTH Action | | | | GQN Mazes | | | | Moving MNIST | | | | Mean |
|---|---|---|---|---|---|---|---|---|---|---|---|---|---|---|---|---|---|
| | SSIM | PSNR | LPIPS | FVD | SSIM | PSNR | LPIPS | FVD | SSIM | PSNR | LPIPS | FVD | SSIM | PSNR | LPIPS | FVD | Rank |
| CW-VAE | 0.65 | 21.20 | 0.31 | 2612 | 0.85 | 26.52 | 0.25 | 2316 | 0.56 | 13.35 | 0.43 | 1276 | 0.64 | 13.03 | 0.26 | 593 | 2.25 |
| NoTmpAbs | 0.56 | 20.39 | 0.36 | 2618 | 0.82 | 25.21 | 0.37 | 2209 | 0.46 | 13.91 | 0.41 | 1210 | 0.66 | 12.93 | 0.26 | 578 | 2.62 |
| RSSM | 0.66 | 23.48 | 0.31 | 2646 | 0.83 | 26.05 | 0.22 | 1915 | 0.42 | 13.56 | 0.45 | 1705 | 0.64 | 12.79 | 0.28 | 589 | 2.62 |
| SVG | 0.56 | 17.22 | 0.32 | 2666 | 0.75 | 21.46 | 0.30 | 1899 | 0.55 | 13.85 | 0.40 | 820 | 0.64 | 11.73 | 0.31 | 1392 | 3.31 |
| VTA | 0.56 | 18.50 | 0.35 | 3305 | 0.77 | 22.41 | 0.24 | 1815 | 0.55 | 13.51 | 0.42 | 1557 | 0.58 | 12.18 | 0.36 | 1274 | 3.62 |

Table E.1: Quantitative comparison of Clockwork VAE (CW-VAE) to state-of-the-art video prediction models across 4 datasets and 4 metrics. The scores are averages across the frames of all evaluation sequences of a dataset. For each dataset and metric, we highlight the best model and those within 5% of its performance in bold. We aggregate the performance by computing the average rank a method achieves across all datasets and metrics (lower is better).

# F    Video Predictions for MineRL Navigate

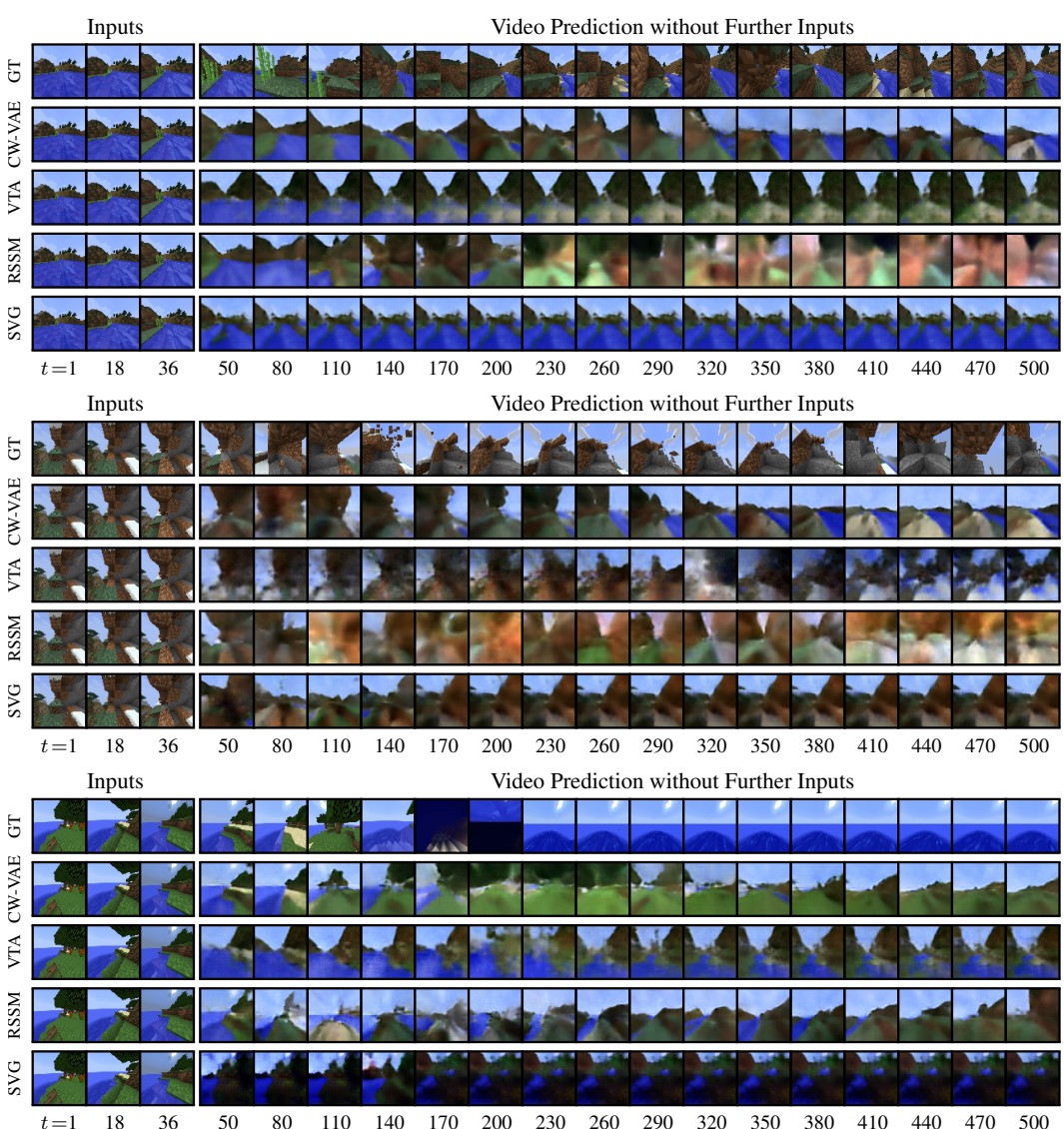

Figure F.1: Additional video predictions on Minecraft.

# G    Video Predictions for 3D Mazes

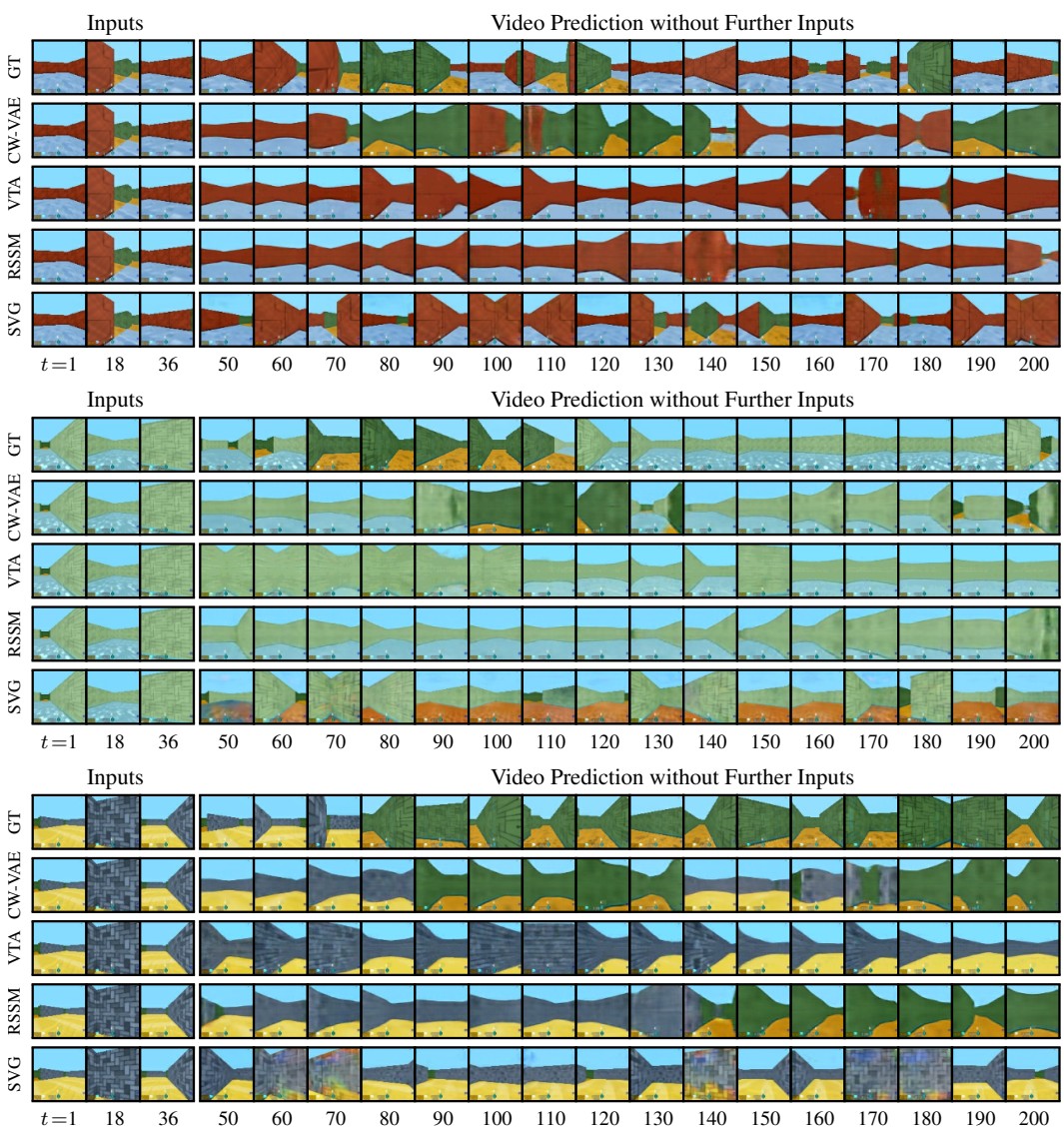

Figure G.1: Additional long horizon open-loop video prediction on GQN.

# H    Top-Level Reset for 3D Mazes

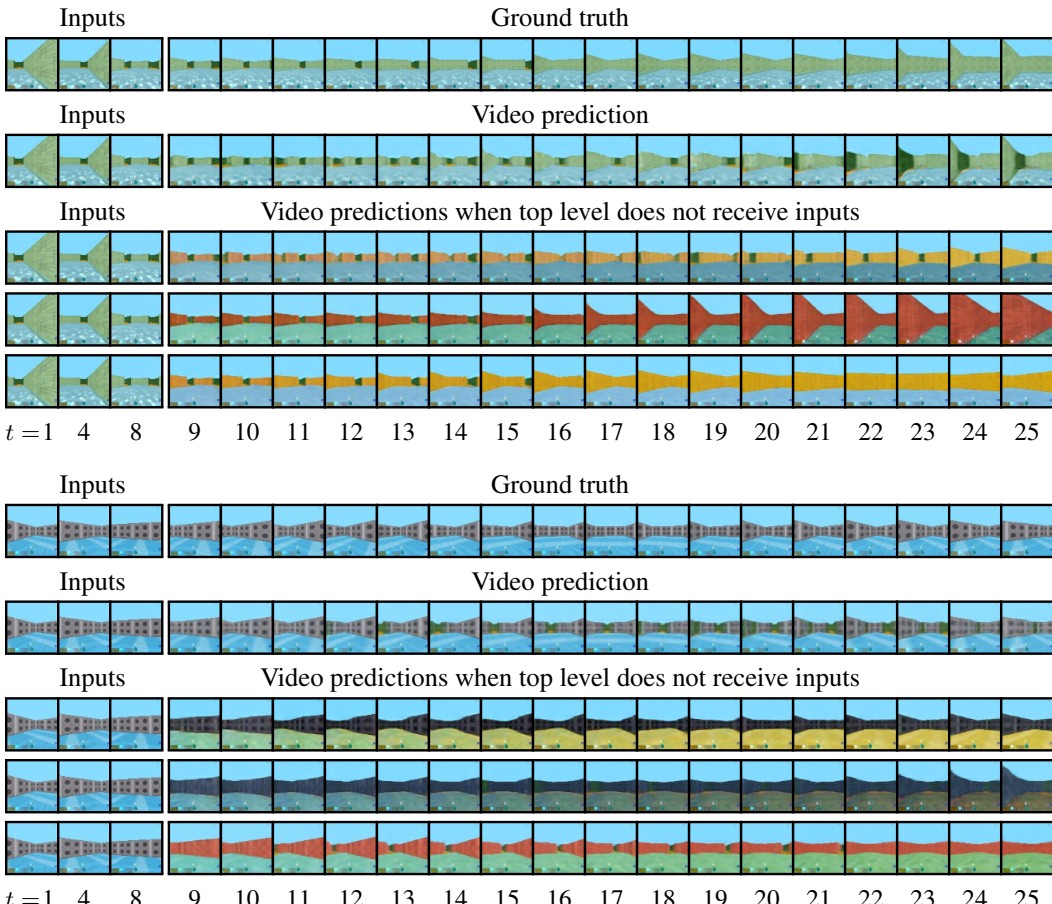

Figure H.1: Additional visualizations of the content represented at different levels of CW-VAE.