# OpenReview forum: "Clockwork Variational Autoencoders"
_NeurIPS.cc/2021/Conference — NeurIPS 2021 Poster_

### Official Review · Reviewer_oxfd · 2021-07-13

**Rating:** 6
**Confidence:** 5

**Summary:**

This paper presents a video prediction model named Clockwork VAE, which leverages hierarchical latent variables and temporal abstraction to learn long-term dependency. It also proposes the new Minecraft benchmark for long-term video prediction. The model is compared with existing video prediction methods on four datasets, and achieves remarkable long-term prediction results.


**Limitations And Societal Impact:**

Yes.

**Main Review:**

Pros:
1. The overall idea of the proposed model is reasonable. It uses a hierarchy of latent variables with a temporally abstract structure to capture long-term video dynamics. It also uses the clockwork recurrent transitions to accommodate different sequence speeds. All the above network designs are integrated in a VAE framework and can be trained in an end-to-end manner.
2. Impressive qualitative results for long-term prediction.

Cons:
1. The idea of combining temporal abstraction, clockwork transition (state copy/update), and variational models has been introduced by VTA. Though a clear difference is that CW-VAE can be equipped with more than two levels of temporal abstraction (that can better capture hierarchical dynamics), it would be good if the authors could further clarify other differences between these two approaches.
2. Another concern of mine is about the completeness of the experiments. It seems that VTA is the most important baseline model, so it would be a good choice if the datasets used by VTA can also be included in this paper. Besides, RSSM is mostly used for model-based RL, where long-term prediction is less important due to the use of re-planning and model predictive control. Since none of the compared approaches were originally designed for long-term prediction, it would be good to include other existing models in the experiments, such as the work from Wichers et al. (2018), which also uses hierarchical latent variables to facilitate long-term prediction.

A minor concern: The authors are encouraged to present the quantitative results of the four metrics and to give more description and analysis in the main text, rather than in the supplementary material. Also, from TableE.1, the performance of the proposed model is not much improved compared with RSSM, especially on the MineRL Navigate dataset.

**Time Spent Reviewing:**

3 hours

---

> ### Author Response · Authors · 2021-08-10
> **Authors' Response**
>
> Thank you for the review and constructive comments! We are happy to see your appreciation of our long-term video prediction results. Below, we address all your concerns: we detail the differences between Clockwork VAE and VTA, our ability to include an evaluation on tasks from the VTA paper, and add further discussion of our quantitative evaluation. We believe that, with these additions, we have addressed the concerns raised in your review. Please let us know if any issues remain!
>
> > *"The idea of combining temporal abstraction, clockwork transition (state copy/update), and variational models has been introduced by VTA. Though a clear difference is that CW-VAE can be equipped with more than two levels of temporal abstraction (that can better capture hierarchical dynamics), it would be good if the authors could further clarify other differences between these two approaches."*
>
> Thank you for this question, which we clarified in the paper. There are a number of differences between Clockwork VAE and VTA:
>
> - As you point out correctly, one difference to VTA is that Clockwork VAE supports any number of levels in the hierarchy.
> - Another difference is that VTA predicts when the low-level should hand control back to the high level, whereas Clockwork VAE uses fixed time scales.
> - Importantly, VTA resets the low-level state when it hands control back to the high level while Clockwork VAE never resets hidden states during a sequence. This prevents the visual inconsistencies found in predictions by VTA that we point out in our paper (Fig 4).
> - Another difference is that VTA uses a smoothing posterior (implemented as a backward encoder RNN) while Clockwork VAE encodes only the images that lie within the windows of the states in the hierarchy.
>
> Empirically, we evaluate Clockwork VAE and VTA on a wider range of more complex datasets than originally presented in the VTA paper, which only uses custom moving dots and maze datasets.
>
> > *"It seems that VTA is the most important baseline model, so it would be a good choice if the datasets used by VTA can also be included in this paper."*
>
> VTA showed experiments on custom datasets of moving dots and mazes. Because those were not made available publicly, we contacted the authors and asked for access to the datasets. However, we have not received a response yet. We look forward to including results on these tasks in case we can secure access to those datasets, ideally still during the discussion period. That said, we believe that the four datasets included in our paper capture the same challenges of the datasets in the VTA paper while going beyond their difficulty in terms of visual complexity and long sequences.
>
> > *"A minor concern: The authors are encouraged to present the quantitative results of the four metrics and to give more description and analysis in the main text, rather than in the supplementary material."*
>
> Thank you for this suggestion. We included the table of numerical results in the appendix due to space constraints, focusing on the aggregated metrics and the semantic evaluation metrics (digit classification and room sequence prediction) in the main text. With the additional page available for accepted papers, we will include the table and a more detailed discussion of it in the main text.
>
> In summary, Clockwork VAE achieves the best average rank across all metrics and datasets. It performs almost at par with RSSM on MineRL Navigate w.r.t. the metrics, but clearly maintains high-level information from the context (such as persistent movement towards an island leads the agent onto the island) over a longer horizon. Clockwork VAE is clearly the best performing model on KTH Action and Moving MNIST due to its ability to maintain high-level object information over a long horizon. We also note that SVG generates very sharp images on GQN Mazes, leading to some high scores with this model. However, our generated video predictions show that SVG struggles with remembering high-level information such as wall and floor patterns over long horizons.

---

> > ### Comment · Reviewer_oxfd · 2021-08-25
> > **Thanks for the response**
> >
> > Thanks for the response! It solves most of my problems. Right now, my only concern is the lack of comparison with other existing models that also focus on long-term prediction. For the current version of this paper, I decided to stick to my score.

---

> > > ### Author Response · Authors · 2021-08-29
> > > **Authors’ Response**
> > >
> > > Dear Reviewer,
> > >
> > > Thank you for confirming that all your concerns have been resolved, except for comparison to other models that also focus on long-term prediction.
> > >
> > > **Comparison with EPVA (Wichers et al., 2018)**
> > >
> > > In terms of conceptual comparison, EPVA is based on separating foreground (that is allowed to change) from background (that is copied from the first input frame). This heuristic does not make sense when the background changes, for example because the camera moves, as in GQN Mazes and MineRL. In contrast, Clockwork VAE does not make any assumptions about the video at the pixel-level but only about the temporal structure inside the abstract latent space.
> > >
> > > Experimentally, a direct comparison between Clockwork VAE and EPVA would be unfair because EPVA relies on encoder/decoder CNNs that were pre-trained on ImageNet (Section 5.2 of Wichers et al.). The adversarial version of their model is also initialized by pre-training. Regardless, we attempted to run EPVA to empirically compare to it, but unfortunately, the authors did not make the pre-trained checkpoint available that the model relies on.
> > >
> > > **Comparison with VTA (Kim et al., 2019)**
> > >
> > > Our paper includes an extensive comparison to the VTA model, a video prediction model specifically designed for long-term prediction that is newer than that by Wichers et al. While Clockwork VAE uses several hierarchical levels with simple fixed timescales, VTA uses only two levels with more complicated prediction of an adaptive time scale.
> > >
> > > Moreover, VTA resets the state at the lower level when the higher level is updated, resulting in low-level inconsistencies in the predictions, as shown in Fig 4 of our paper (around t=55). Experimentally, VTA outperforms the non-hierarchical model SVG but Clockwork VAE clearly outperforms both of them on all 4 datasets we evaluted on.
> > >
> > > If there is another video prediction that we should compare to in our paper, we to ask you to mention the specific model and we will be happy to provide such a comparison.

---

> > > > ### Comment · Reviewer_oxfd · 2021-08-30
> > > > **I raised my score**
> > > >
> > > > Thanks to the authors for further discussion of the relationship between these models. I think overall this is a very good paper with an interesting idea and impressive long-term prediction performance. I raised my score accordingly.

---

> > ### Author Response · Authors · 2021-08-25
> > **Authors' Response**
> >
> > In addition to our response above, we attempted to run the model by Wichers et al. as you suggested. However, we noticed that their video prediction model relies on pre-trained encoder and decoder models, and the weights for these models were not made available by Wichers et al., making it difficult to replicate their model.

---

### Official Review · Reviewer_UP8B · 2021-07-13

**Rating:** 7
**Confidence:** 4

**Summary:**

This paper proposes Clockwork-VAEs, a hierarchical latent dynamics video prediction model. CW-VAE uses different clock speeds at each level to model dependencies occurring at different frequencies. This allows it to perform well on long-term video prediction. The paper presents results on four diverse datasets, including using a new dataset (MineRL) for long term video prediction.


**Limitations And Societal Impact:**

Limitations of CW-VAE and potential directions for future work are provided in Section 4 of the main paper. There does not seem to be any potential negative societal impact that would directly arise from this work.

**Main Review:**

The proposed architecture is intuitive, simple, and achieve strong results. Experiments are also fairly comprehensive, and I appreciate the visualizations provided. I think this model and the proposed use of the MineRL dataset are valuable contributions for the community, and I lean towards acceptance.

## Strengths

* The proposed CW-VAE automatically adapts to the frame rate of the dataset, which is a neat feature to sidestep annoying hyperparameter tuning across different datasets.
* The quantitative results are strong, and the ablation experiments are comprehensive. I particularly appreciate the efforts in sections 3.6, 3.8 and 3.9 which examine different levels of CW-VAE, to show that the proposed hierarchy does indeed encode information of differing content and temporal speeds.
* The model itself is intuitive and simple, yet appears to achieve strong results. Training time and resources are also modest and comparable to prior work.


## Weaknesses

* The experiments seem to be conducted on fairly simple datasets, which have minimal moving objects. The only dataset with more than one moving object is Moving MNIST, which is somewhat of a toy dataset. Experiments on more complex datasets such as KITTI would be valuable. It would be interesting to see how the different hierarchies perform with faster / slower moving cars, in particular, or how it operates when there are multiple moving objects in the video.
* While the quantitative metrics seem strong, it is also valuable to conduct a human study over the generated results, especially for long term generation results on MineRL, which is a new dataset.
* There is missing comparison to prior work on stronger hierarchical models, such as [1].
* The results in the paper are presented on 64x64 images, which seem rather small. How does CW-VAE perform if scaled up to larger video resolutions, such as 128x128 or 256x256?

## Concerns / questions

* Are results presented in Figures F.1 and G.1 randomly sampled, or cherry picked? If they are cherry picked, please also share some randomly sampled results.
* How is the video diversity of CW-VAE generated videos? There is a claim in the paper (L198) that VTA lacks diversity on MineRL. Please substantiate CW-VAE’s improved diversity with qualitative results for the different models, or with some measure of video diversity (e.g. LPIPS of predicted video frames at long horizons) when sampling different values from the Gaussian prior.
* MineRL seems like a natural setting for action-conditioned video prediction. It would be interesting to hear the authors’ thoughts on whether they attempted to include the action condition in CW-VAE, or if there is a natural way to do so.

## Presentation

There are some minor typos in the paper:
* L169: “factors training faster” -> “factors *train* faster”


[1] Lee, W., Jung, W., Zhang, H., Chen, T., Koh, J. Y., Huang, T., ... & Hong, S. (2020, September). Revisiting Hierarchical Approach for Persistent Long-Term Video Prediction. In International Conference on Learning Representations.


*Post rebuttal*

* The authors have addressed most of my concerns. I think this is a meaningful paper and I would be happy to accept it. I have raised my score accordingly.

**Time Spent Reviewing:**

4

---

> ### Author Response · Authors · 2021-08-10
> **Authors' Response**
>
> Thank you for the review and constructive comments! Below, we address each of your raised points: we highlight our focus on temporal rather than visual complexity for this work, discuss the recent paper on long-term prediction you mentioned, and add additional samples of Clockwork VAE. If this does not address all of your questions or concerns, please let us know!
>
>
> > *"Experiments on more complex datasets such as KITTI would be valuable. It would be interesting to see how the different hierarchies perform with faster / slower moving cars, in particular, or how it operates when there are multiple moving objects in the video."*
>
> In this work, we focused on learning long-term temporal dependencies in video, and we chose datasets that would demonstrate improvements in this direction. We agree that analyses of the different hierarchies with real-world car driving videos with multiple moving objects would be interesting. However, we recognize scaling to such datasets with higher visual complexity as an orthogonal research problem, while being confident that future solutions in this direction (such as more powerful encoder/decoder architectures) will be compatible with Clockwork VAE. We To this end, we also highlight that the previous state-of-the-art method for long-term video prediction using latent hierarchies, VTA, evaluates on even simpler datasets (custom moving dots and maze datasets). In comparison, our paper moves in the right direction towards more complex datasets, such as MineRL.
>
> > *"While the quantitative metrics seem strong, it is also valuable to conduct a human study over the generated results, especially for long term generation results on MineRL, which is a new dataset."*
>
> Thank you for the suggestion. We like the idea of including a human evaluation of the models in our paper. Because this will take some time, we are working to include this in the final version of the paper. From observing the open-loop video predictions ourselves, we find that Clockwork VAE clearly outperforms the other methods.
>
>
> > *"There is missing comparison to prior work on stronger hierarchical models, such as [1]."*
>
> Thank you for bringing this recent work to our attention. While Clockwork VAE learns abstract latent states end-to-end and could in principle be applied to sequences of any modality, the model proposed by Lee et al. contains several video-specific priors and requires additional supervision in the form of segmentation masks. Specifically, their predictions are conditioned on pixel-level segmentation masks predicted by another model that is trained separately using additional supervision, and explicitly incorporate the notion of pixel-level occlusion and optical flow into their model. Their model does not use any temporally abstraction structure. For applications in video prediction, it would be interesting to combine these orthogonal ideas with the contributions of Clockwork VAEs.
>
>
> > *"The results in the paper are presented on 64x64 images, which seem rather small. How does CW-VAE perform if scaled up to larger video resolutions, such as 128x128 or 256x256?"*
>
> We focused our experiments on images of resolution 64x64 as this allows us to use the same encoder/decoder architecture across all tasks. That said, Clockwork VAE is compatible with any encoder/decoder architecture (e.g. DCGAN, VGG, ResNet, ViT) and thus is applicable to datasets with higher resolutions as well. We did not run our model on larger resolutions, because it seemed orthogonal to our goal of evaluating the ability to learn long temporal dependencies and would have increased the computational requirements.
>
> > *"Are results presented in Figures F.1 and G.1 randomly sampled, or cherry picked? If they are cherry picked, please also share some randomly sampled results."*
>
>
> Because we cannot show all frames of the long video sequences in the figure of the paper, we selected representative predictions out of the first 20 sequences in the dataset for which the interesting transitions are visible in the figure (that shows only 1 every K frames). We added random samples for the MineRL and GQN Mazes datasets. We also added these links to the paper and do the same for the remaining datasets.
>
> [MineRL random samples](https://drive.google.com/file/d/17aLKi84L_wJUUeGwdQv3e8kjKkIxDC8E/view?usp=sharing)
>
> [GQN Mazes random samples](https://drive.google.com/file/d/1-pSybTqKWjh_PyLJ1-Hb3z4ROSjRIa2i/view?usp=sharing)
>
> > *"MineRL seems like a natural setting for action-conditioned video prediction. It would be interesting to hear the authors’ thoughts on whether they attempted to include the action condition in CW-VAE, or if there is a natural way to do so."*
>
> To keep a simple experimental setup, we did not use action-conditioning in our experiments. We agree that MineRL is a natural candidate for this. We released the MineRL dataset together with the action information to allow the community to use the dataset either with and without actions. For Clockwork VAE specifically, these are several options to condition on actions: (1) one could condition each level on the sequence of K actions it corresponds to, (2) one could condition only the lowest level on actions, leveraging the higher levels to learn the unconditional prior, or (3) learn abstract actions through skill discovery or sequence clustering to condition the higher levels. We think this is a very interesting direction for future work that can bring Clockwork VAEs closer to applications in reinforcement learning.
>
> [1] Lee et al. Revisiting Hierarchical Approach for Persistent Long-Term Video Prediction. ICLR 2021. https://arxiv.org/pdf/2104.06697.pdf

---

> > ### Comment · Reviewer_UP8B · 2021-08-23
> > **Rebuttal comment**
> >
> > Thank you for the comprehensive discussion! I generally agree that hierarchical video generation is interesting and somewhat of an orthogonal direction relevant for future work.
> >
> > I have one concern that was not addressed, regarding the diversity of CW-VAE generated videos. In your paper (L198), it's claimed that VTA lacks diversity on MineRL. Can you provide some details regarding how CW-VAE fares w.r.t. diverse generations from the same initial conditions?

---

> > > ### Author Response · Authors · 2021-08-26
> > > **Authors' Response**
> > >
> > > Thank you for your response! With regards to our comment on the lack of diversity in video generations from VTA, we meant to point out the lack of different modalities in the scenes generated by this model, when compared to those generated by Clockwork VAE. We found this to be even clearer with the GQN Mazes Dataset, and have updated the paper, and will update our project webpage in the future, with this analysis. We would like to illustrate this with some random examples from the GQN Mazes dataset, which can be found on [this link](https://drive.google.com/file/d/1tne0D5AYf3RsNYrm1yjKI-mm0tialz3E/view?usp=sharing). Only in 2 out of 20 examples do we see that VTA generates video outside of the room it started in, while Clockwork VAE generated different rooms/hallways in 19 out of 20 examples. It is in this context that we stated that generations from VTA lack diversity.
> > >
> > > We hope that this addresses your concern. Please let us know if there is anything else that you would like to discuss, and we would be happy to take it up.

---

> > > > ### Comment · Reviewer_UP8B · 2021-08-26
> > > > **Rebuttal comment**
> > > >
> > > > Thanks for providing this, it addresses my concerns and I am happy to raise my score.

---

### Official Review · Reviewer_ugeK · 2021-07-16

**Rating:** 6
**Confidence:** 4

**Summary:**

In this paper, the authors tackle the problem of long-term video generation. To this end, they propose a hierarchical latent variational autoencoder: The Clockwork Variation Autoencoder. This model incorporates a hierarchy of latents separated at different frame rates. This frameworks allows the VAE to model dependencies in the video at different timescales. The authors evaluate their approach on a variety of datasets: a new benchmark from Minecraft, KTH, Moving MNIST, and GQN Mazes. The paper presents promising qualitative and quantitative results with the proposed approach.

**Ethical Concerns:**

I see no ethical issues with this paper.

**Limitations And Societal Impact:**

The authors have addressed any limitations or social impact.

**Main Review:**

Strengths:

1. The paper is well written and easy to understand. It flows well and properly structured.

2. The problem is relevant and important to the community: There has been a cornucopia of video modeling papers, but relatively few have addressed the problem of long-term coherence in video generation. This is a fairly unexplored problem relevant to other fields such as Reinforcement Learning.

3. The quantitative and qualitative results on the chosen datasets is promising.

Weaknesses:

1. Choice of metrics. This is an issue common with video modeling papers in general, but metrics such as SSIM, PSNR, and
LPIPS assume a single ground truth future video. They assume the problem of future video generation is a deterministic problem
when more stochastic enviroments (i.e. real-world videos from UCF101, Kinetics, etc) clearly have multiple valid outcomes. Video
prediction should be treated as a generative model that estimates the probability distribution of future video. Metrics such as Log-likelihood (or approximations to this), and even crowdsourced human ratings would set a precedent for future work on more complex datasets.

2. The datasets are simplistic. From a computer vision standpoint, the datasets featured have very low visual complexity.
From an event standpoint, KTH, Moving MNIST, and GQN Mazes are domains with very low levels of uncertainty.


Overall:

The paper has a few flaws. The authors choose simple datasets for evaluation and use flawed metrics that may not be the best foundation for future work. However, they explore an important, underexplored problem in video generation. Modeling long-term dependencies in many domains-text, images, and video-is non-trivial. In spite of these issues, I believe this paper is a step in the right direction for research in video generation.


**Time Spent Reviewing:**

1 Hour

---

> ### Author Response · Authors · 2021-08-10
> **Authors' Response**
>
> Thank you for the review and constructive comments! Below, we address each of your raised points: we report ELBO numbers for the models on one MineRL and explain why these do not offer a meaningful comparison, we agree that scaling to tasks of higher visual complexity is an interesting future direction and discuss why we see Clockwork VAE as an independent and important contribution, and we clarify the stochastic nature of the GQN Mazes dataset. If this does not address all of your questions or concerns, please let us know!
>
> > *"Choice of metrics. This is an issue common with video modeling papers in general, but metrics such as SSIM, PSNR, and LPIPS assume a single ground truth future video. Video prediction should be treated as a generative model that estimates the probability distribution of future video. Metrics such as Log-likelihood (or approximations to this), and even crowdsourced human ratings would set a precedent for future work on more complex datasets."*
>
> We agree that common metrics in video prediction are not suitable for evaluating predictions on datasets with many plausible futures, which is exaggerated for long-horizon predictions. This motivated our use of more semantically meaningful evaluation metrics in this work, such as classifying the digits of the predictions for Moving MNIST and comparing the room sequences in the model predictions for GQN Mazes. As requested, we also report the test ELBO (for Clockwork VAE and VTA) and the MSE and KL loss (for SVG) on the MineRL dataset:
>
>
> | Model | ELBO (nats) |
> | :---- | ----------: |
> | CW-VAE | -209.6 |
> | VTA | -836.4 |
> | SVG* | -39100.6 |
>
> (* SVG is trained using MSE and KL losses, which are 24.1 and 39074.7 on the test set, respectively. This could be converted to an ELBO score under the assumption of a Gaussian decoder with standard deviation of sqrt(0.5). For this, one first converts the MSE into a conditional log likelihood: LL = -MSE / 2sqrt(0.5) - sqrt(2*pi*sqrt(0.5)^2) = MSE / 1 - sqrt(pi) = -24.1 - 1.8 = -25.9. Subtracting the KL yields an ELBO of -39100.6.)
>
> In practice, evaluation by ELBO is known to suffer from being heavily dependent on the decoder standard deviation [1]. This can lead to large differences in ELBO. For example, before we tuned the decoder standard deviation of VTA, it achieved an ELBO of -11424.443. After tuning its decoder standard deviation from 1 to 0.4, the ELBO becomes -836.377 even though the model's predictions look very similar in both cases. We thus conclude that comparison by ELBO is not as meaningful as we hoped and instead refer to our semantic evaluation metrics for reliable comparison.
>
> > *"From a computer vision standpoint, the datasets featured have very low visual complexity."*
>
> We agree that the datasets are not as visually complex as in some video prediction papers that focus more on generating sharp images (e.g. using adversarial losses). However, this is not the focus of our paper. Instead of evaluating the ability to learn spatial dependencies between pixels within individual frames, our evaluation targets the ability to learn long temporal dependencies. Clockwork VAE clearly outperforms prior methods in this regard. Moreover, we highlight that the previous state-of-the-art method for long-term video prediction using latent hierarchies, VTA, evaluates on even simpler datasets (custom moving dots and maze datasets). In comparison, our paper moves in the right direction towards more complex datasets, such as MineRL.
>
> That said, Clockwork VAE is compatible with any encoder/decoder architecture, so that larger architectures (ResNet, ViT) can be used for tasks of higher visual complexity. We see this as an orthogonal direction and thus do not investigate it in this paper, but think of it as a promising research direction.
>
> > *"From an event standpoint, KTH, Moving MNIST, and GQN Mazes are domains with very low levels of uncertainty."*
>
> MineRL and GQN Mazes both feature procedurally generated environments. GQN Mazes, in particular, consists of sequences generated from randomly generated mazes where the agent is initialized at random start positions in every sequence. Thus, the two datasets include a large amount of inherent stochasticity and our probabilistic model is well-equipped to learn. If this is not what you referred to by "event standpoint", we would like to ask you to clarify further.
>
> [1]: Lucas et al. Don’t Blame the ELBO! A Linear VAE Perspective on Posterior Collapse. NeurIPS 2019. https://arxiv.org/pdf/1911.02469.pdf

---

> > ### Author Response · Authors · 2021-08-28
> > **Recap of Authors’ Response**
> >
> > Dear Reviewer,
> >
> > We would be grateful if you can confirm whether our response addressed all your concerns, and let us know if any issues remain. To recap our response, we:
> >
> > - Reported ELBO numbers for the models on MineRL
> > - Discussed why a comparison by ELBO is unfortunately not meaningful
> > - Motivated the use of our semantically meaningful evaluation metrics
> > - Discussed our motivation behind choosing the datasets we evaluated on
> > - Discussed the scalability of Clockwork VAE to tasks of higher visual complexity
> > - Clarified the stochastic nature of the MineRL and GQN Mazes datasets

---

> > > ### Comment · Reviewer_ugeK · 2021-08-28
> > > **Response to Recap**
> > >
> > > Yes, your response has addressed my concerns. I stand by my rating of accept and agree with many of the other reviewers' positive assessments of the paper.

---

### Official Review · Reviewer_qetJ · 2021-07-16

**Rating:** 7
**Confidence:** 4

**Summary:**

The paper proposed a method for video prediction called Clockwork Variational Autoencoders. The method adapts Clockwork-RNNs to sequential VAEs and defines a hierarchy of latents operating at different temporal scales. This allows the model to generate long term videos.

**Ethical Concerns:**

There are no major ethical concerns with the paper. The authors use a dataset showing videos of humans but do not mention the license for this dataset (it might be unclear), as opposed to the rest of the dataset for which they mention the license.

**Limitations And Societal Impact:**

The authors discuss some of the limitations of their model but do not acknowledge any potential negative societal impact of their work. Video prediction models in general might be used with malicious intents (for example "deepfakes"), and might reflect biases found in the training data. Please include a discussion on potential negative impacts.

**Main Review:**

**Originality**

The idea of having different temporal scales in sequential models has been proposed before in Chung et al. or Koutnik et al. However, the adaptation of this idea to sequential VAEs for video prediction is not straightforward and therefore novel.

**Clarity**

The paper is well written. The problem statement, the model description and the experimental section are clear.

**Quality**

The technical contribution of the paper is sound. The experimental section of the paper is extensive and adequately highlights the properties of the proposed model, not only comparing to different baselines but also investigating different aspects of the model such as the utilization of the different levels of latents.

**Significance**

Overall the paper is clearly written, presents a novel approach for video prediction specifically designed to model long-term videos, and the experimental section properly supports the validity of the model. I believe the community will be interested in this work.

**Review Updates**

After reading the rest of the reviews and the authors' response, I stand by my original rating and argue for the acceptance of the paper.

**Time Spent Reviewing:**

3

---

> ### Author Response · Authors · 2021-08-10
> **Authors' Response**
>
> Thank you for the review and constructive comments!
>
> > *"The authors discuss some of the limitations of their model but do not acknowledge any potential negative societal impact of their work."*
>
>
> Thank you for bringing this to our attention. We added a paragraph to the paper to discuss the potential societal impact of our work. In summary, video prediction models using latent variables (in our case of different temporal resolution) could be used to alter certain semantic aspects of a video while leaving others intact. This could be used to create fake videos with the intent to deceive people and spread misinformation. Additionally, use of such prediction models for video surveillance is sensitive to inherent biases in training data and may lead to unwanted discriminations in the society. On the other hand, this also allows for semantic editing to accelerate movie production and serve as a creative tool for artists.
>
> > *"The authors use a dataset showing videos of humans but do not mention the license for this dataset (it might be unclear), as opposed to the rest of the dataset for which they mention the license."*
>
> Thank you for pointing this out. The KTH action dataset is available under the CC Attribution-NonCommercial license and we added this information to our paper.

---

### Decision · Program_Chairs · 2021-09-27

**Decision:**

Accept (Poster)

**Comment:**

This paper proposes a novel architecture for long term video generation. Reviewers agree that the approach is novel, it focuses on an important and under-explore problem of video generation and that the experimental section supports the value of the contribution.